# Assessment of Waterbird Habitat Importance and Identification of Conservation Gaps in Anhui Province

**DOI:** 10.3390/ani14071004

**Published:** 2024-03-25

**Authors:** Yuan Liu, Xianglin Ji, Lizhi Zhou

**Affiliations:** 1School of Resources and Environmental Engineering, Anhui University, Hefei 230601, China; x21301088@stu.ahu.edu.cn (Y.L.); jixianglin@stu.ahu.edu.cn (X.J.); 2Anhui Province Key Laboratory of Wetland Ecosystem Protection and Restoration, Anhui University, Hefei 230601, China; 3Anhui Shengjin Lake Wetland Ecology National Long-Term Scientific Research Base, Chizhou 247230, China

**Keywords:** waterbirds, critical habitat, habitat assessment, gap analysis, conservation management

## Abstract

**Simple Summary:**

Wetlands are one of the most important habitats for waterbirds. Due to the increase in human activities, waterbird habitats have been severely disturbed. Limited availability of human and natural resources makes effective habitat protection particularly important. Therefore, in this study, we developed an index system to evaluate habitat waterbird importance in Anhui Province based on habitat status. This allowed us to identify distribution and change patterns over time in important waterbird habitats. We also examined gaps in the protection of these crucial habitats and identified the key environmental influences. We found that Anhui Province has 73 important waterbird habitats, mainly concentrated in the Yangtze River floodplain. Gap analysis showed that 42 of these suffered from protection gaps, accounting for 57.53% of the total area. The importance of waterbird habitats was significantly correlated with elevation, normalized vegetation index, lake area, and lake circumference but not with distance from roads or population density. This study presents scientific information for waterbird habitat conservation in Anhui Province and provides reference cases for waterbird habitat conservation and maintaining wetland ecosystem functions.

**Abstract:**

Wetlands are among the most important habitats of highly wetland-dependent waterbirds but are subject to ongoing habitat loss and degradation owing to intensified anthropogenic activities. The scarcity of human and natural resources makes effective habitat protection an important concern. Here, we aimed to investigate waterbird habitat protection methods for Anhui Province, China, a critical stopover and wintering area on the East Asian-Australasian Flyway that features rich wetland resources subject to significant habitat loss and degradation. We evaluated the status and importance of 306 wintering waterbird habitats and identified the key environmental influences and current protection gaps using the entropy weights method and generalized additive modeling. We found 73 important habitats for waterbirds in Anhui Province, which were classified into levels of importance (descending from I to V) according to the natural discontinuity method. Level I and Level II habitats were mainly located in the Yangtze River floodplain and Level IV habitats in the Huaihe River floodplain. The gap analysis showed that 42 important waterbird habitats had protection gaps, accounting for 57.53% of the total area. Waterbird habitat importance was significantly correlated with elevation, normalized vegetation index, lake area, and lake circumference but not with distance from roads or population density. The results of this study provide scientific information for waterbird habitat conservation planning, which is crucial for maintaining wetland ecosystem functions.

## 1. Introduction

Wetlands are important as habitats for wildlife and as part of natural ecosystems [1]. Different wetland habitat types attract different fauna assemblages [2]. Waterbirds as a highly wetland-dependent group depend strongly on wetland habitat quality, which can also affect the ability to maintain wildlife diversity [3]. However, wetland ecosystems are under ongoing pressure due to accelerating economic and social development as well as intensifying global climate change and human activities. Consequently, the structure of wetland ecosystems has functionally changed, and habitat changes have reduced biodiversity [4,5,6]. Pressures on waterbird survival have increased. As a result, species diversity has suffered [7].

The establishment and maintenance of protected area systems is central to regional and global waterbird conservation strategies that aim to safeguard waterbird populations against further decline [8]. The management of important waterbird habitats is, therefore, essential for the conservation of waterbird diversity and protecting wetland ecosystems. However, effective habitat conservation is challenging because of limited human and natural resources [9], which makes improvement of the conservation efficiency of habitats particularly important. To this end, habitat importance assessment and conservation gap analysis constitute applicable methods [10].

Waterbird habitat importance can be determined based on measures of community composition and structure, such as species abundance and richness. Populations of endangered and nationally protected waterbirds, such as cranes, storks and pelicans, tend to be particularly vulnerable, and their survival rates are highly dependent on habitat quality [11]. The population status of such species can, thus, be employed as an indicator in habitat importance assessment. As a number of environmental factors directly affect the survival strategies of waterbirds and determine their habitat selection [12], an understanding of the relationship between these factors and habitat importance is fundamental for devising effective management strategies [13]. A further important conservation approach is to identify protection gap areas by overlaying the distributions of important habitats and protected areas using spatial analysis tools [14,15].

Anhui Province in China is an important area for waterbirds on the East Asian–Australasian Flyway because of its amenable climate and richness in wetland resources of different types, including reservoirs, rivers, and lakes. Many waterbirds, including endangered species, such as the Siberian Crane (*Grus leucogeranus*), Oriental Stork (*Ciconia boyciana*), and Baer’s Pochard (*Aythya baeri*), regularly use the region as a winter stopover [16,17,18]. Anhui Province has prioritized the protection of wetlands by continuously strengthening the protection of waterbirds and their habitats; however, the protection system requires improvements because there remain unprotected or insufficiently protected waterbird habitats and protection gaps. Some waterbird habitats have also been destroyed by anthropogenic ecological pressures, such as reclamation, aquaculture, grazing, road construction, and dam impoundment [7]. Analyses of important waterbird habitats and conservation gaps in the region are, therefore, urgently required.

This study explores scientific approaches to waterbird habitat conservation, taking Anhui Province as a case study. Data obtained from waterbird surveys were used to determine their distributions, and the distribution patterns of important waterbird habitats were determined through importance assessment and overlapped with established protected areas to further assess current protection levels and identify protection gaps. The three research aims were to: (1) obtain distribution patterns of waterbirds and their temporal changes in Anhui Province, create a ranked list of important regional waterbird habitats, and generate habitat distribution maps; (2) conduct a gap analysis using GIS to clarify protection gaps and generate habitat protection gap maps; and (3) identify the environmental factors that affect habitat importance. The results of the study are anticipated to provide information for waterbird habitat conservation management and can be expanded to other regions.

## 2. Materials and Methods

### 2.1. Study Area

Anhui Province is rich in wetland resources, which are mainly distributed in the floodplains of the Yangtze, Huaihe, and Xin’an River (Figure 1). Present wetland types principally consist of rivers, lakes, marshes, and artificial wetlands. The Yangtze River floodplain mainly consists of shallow river-connected lake wetlands; those of the Huaihe River, of reservoir type wetlands; and those of the Xin’an River, of riverine wetlands. In spring and summer (wet season), aquatic plants grow strongly and accumulate rich nutrients, while in autumn and winter (dry season), the mudflats are gradually exposed, providing rich food resources for wintering waterbirds. The suitable habitat and abundant food resources make Anhui Province an important stopover and wintering location for migratory waterbirds.

### 2.2. Data Sources and Processing

We selected 306 wetland units in the Yangtze, Huaihe, and Xin’an River floodplains, each of which was surveyed four times: July 2022 (breeding season); November 2022 (autumn migration); January 2023 (wintering); and March 2023 (spring migration). Each wetland unit (one lake or one river) was surveyed in a single day. Each floodplain was completed in 7–10 days to avoid the confounding effects of migratory movements. Each survey was conducted by 25–30 people, and individual types of wetland units were surveyed by fixed personnel to provide surveyor consistency. Surveys used the point count and line counting methods, with each survey point constituting a semicircle with a radius of approximately 500 m. The total observed coverage area of the sample points in each lake, river, or reservoir was at least 70% of the total area. The total number of waterbirds in each unit was estimated by multiplying density by area. Binoculars (8.5 × 42; Swarovski, Absam, Austria) and monoculars (ATM 20–60 × 85; Swarovski) were used for identification and counting. Direct counting was performed in areas with few waterbirds and group counting in areas with large bird assemblages [19,20]. The survey was conducted in clear, windless weather at 3 h after sunrise and at 3 h before sunset, in summer, spring, and autumn, with observations lasting approximately 15–20 min. Six types of data were selected as the factors potentially influencing habitat importance. The habitat area and perimeter were obtained using ArcGIS 10.7 (ESRI, Redlands, CA, USA). Habitat elevation (DEM) data were obtained from the Geospatial Data Cloud Platform of the Computer Network Information Center of the Chinese Academy of Sciences (http://www.gscloud.cn, accessed on 15 April 2023). Vegetation coverage (normalized vegetation index [NDVI]) and population density data were obtained from the Resources and Environmental Sciences and Data Center of the Chinese Academy of Sciences (https://www.resdc.cn, accessed on 16 April 2023). Habitat distance to roads was obtained from OpenStreetMap.

### 2.3. Statistical Analysis

#### 2.3.1. Habitat Importance Assessment for Waterbirds

An evaluation index system of waterbird habitat importance in Anhui Province was constructed based on three aspects: diversity, importance, and suitability (Table 1). Seventeen indicators were used to assess these aspects, using information based on surveyed species abundance and richness, International Union for Conservation of Nature (IUCN) Red List of Threatened Species data, and national key protected species data. The entropy weights method was used to determine the weights of the indicators, resulting in an importance score for waterbird habitat for each survey period, which were summed for the total score. We then used this score to classify habitats into five levels of importance, based on the natural discontinuity grading method [21], ranging from level I (most important) to level V (least important).

The entropy weights method is an unbiased and objective information weighting model used to optimize decision making based on qualitative, quantitative, or conflicting factors [21]. It has become the main method for assessing habitat importance while avoiding the inherent subjectivity generated by human judgment by weighting indices using the natural distribution of indices [22].

#### 2.3.2. Protection Gap Analysis

Gap analysis is a method for determining the suitability of existing biodiversity conservation systems and identifying unprotected or under-protected biodiversity components [23]. It has been widely used for the conservation of species diversity in plants [24], birds [25], amphibians [26], and endangered species [27], along with playing an important role in single species conservation [28]. We employed the method to create distribution maps of important waterbird habitats and of protected areas in Anhui Province. The superposition of spatial data and calculation of attribute data were used to identify important habitats outside the current protection range as protection gaps and characterize them [29]. In our study, we delineate wetland protection areas in Anhui Province, drawing on both national and provincial nature reserves and parks. By overlaying spatial data and analyzing attribute data for important waterbird habitats and existing wetland protection areas in Anhui Province, we identify gaps in wetland protection.

#### 2.3.3. Analysis of Environmental Factors of Important Habitats

Generalized additive models (GAMs) can be used to explain highly nonlinear or non-monotonic relationships between response variables and environmental factors [30]. The flexibility of the fitted model can help explain complex relationships and assist in elucidating the importance of each explanatory variable. We used GAMs to model the relationship between environmental factors and waterbird habitat importance. Six environmental factors were selected: rural population density, habitat elevation (DEM), NDVI, distance from the road, lake area, and lake perimeter. Modeling was performed using the “gam” function in the “mgcv” package in R 4.1.3 software.

## 3. Results

### 3.1. Distribution Pattern of Waterbirds

From 2022 to 2023, we recorded 1,460,875 waterbirds in Anhui Province (Figure 2). These covered 100 species from eight orders and 18 families. Of these species, 27 were national key protected species (12 first level and 15 s level species). Additionally, two globally critically endangered (CR) species (*Aythya baeri* and *Grus leucogeranus*), three endangered (EN) species (*Mergus squamatus*, *Ciconia boyciana*, and *Platalea minor*), nine vulnerable (VU) species, and seven near threatened (NT) species were found.

Surveys in the four periods showed that waterbird abundance and richness were highest in the wintering period, followed by the autumn migration period, the spring migration period, and the breeding period. National Key Protected Species abundance was highest during wintering and lowest during breeding. The abundance of CR species was highest during autumn migration, followed by spring migration, wintering, and breeding. EN, VU, and NT species abundance was highest during wintering, followed by spring migration, autumn migration, and breeding (Table 2). One must note that the spring survey migration period was longer than the autumn migration period.

Waterbird survey data were further analyzed separately for the three major floodplains in Anhui Province. Abundances at all levels were highest in the Yangtze River floodplain, followed by in the Huaihe and Xin’an River floodplains (Table 3).

### 3.2. Distribution Pattern of Important Habitats for Waterbirds

Based on the survey data from the four periods, the importance of waterbird habitats in Anhui Province was assessed. A distribution map of important habitats in each period was generated (Figure 3). During wintering, 47 important habitats were identified with importance scores ranging from 0.2779 to 7.2198. Three Level I habitats were located at Shengjin (7.2198), Caizi (6.6742) and Huanghu Lake (4.9567). Additionally, 7 Level II habitats, 13 Level III habitats, and 24 Level IV habitats were identified. During spring migration, 43 important waterbird habitats were found with scores ranging from 0.2689 to 8.6119 and consisted of one Level I habitat located at Shengjin Lake (8.6119), 8 Level II habitats, 17 Level III habitats, and 17 Level IV habitats. During breeding, 41 important waterbird habitats were identified with scores ranging from 0.2044 to 8.2398. Among these habitats, one Level I habitat was located at Huangpi Lake (8.2398), along with four Level II habitats, six Level III habitats, and thirty Level IV habitats. Finally, during autumn migration, 36 important waterbird habitats were identified with scores ranging from 0.3140 to 7.2198. These habitats consisted of 2 Level I habitats located at Shengjin Lake (7.6657) and Wuchang Lake (6.3113), as well as 5 Level II habitats, 11 Level III habitats, and 18 Level IV habitats.

Importance scores summed across the duration of the entire study are shown in Figure 4. In total, 73 important waterbird habitats were found with importance scores ranging from 0.0400 to 24.7745. Score ranges per level were as follows: Level I, 24.7745–13.3592; Level II, 13.3591–5.5962; Level III, 5.5961–1.7576; Level IV, 1.7576–0.5134; Level V 0.5134–0.0400 (Appendix A).

Scores for Levels I–IV ranged from 0.5342 to 24.7745 (Table 2). Level I habitats included Shengjine, Huangpi, Wuchang, and Caizi Lake, all of which are located in the Yangtze River floodplain. The six Level II important waterbird habitats identified include Huanghu, Daguan, Longguan, Chenyao, Po Lake, and Luxi River. The Luxi River belongs to the Huaihe River floodplain, while the other locations belong to the Yangtze floodplain. Of the 18 Level III habitats, 12 were located in the Huaihe River floodplain and 6 in the Yangtze floodplain. Of the 45 Level IV habitats, 23 were located in the Huaihe River floodplain, 20 in the Yangtze River floodplain, and 2 in the Xin’an River floodplain (i.e., the Hengjiang and Changjiang Rivers).

### 3.3. Protection Status and Gaps

A spatial overlay of important waterbird habitats and existing protected areas in Anhui Province was performed using ArcGIS to determine areas of protection gaps. A total of 42 (57.53%) of the 73 identified important waterbird habitats had no established protected areas (Figure 5). The degree of protection gaps varied by region with fewer protection gaps and higher importance indices, such as those of Huangpi and Chaohu Lake, Wanhe River, and Nanyi Lake, in the Yangtze River floodplain; in contrast, more protection gaps but generally lower importance indices, such as those of Qili Lake, Dongpi River, and Wabu Lake, in the Huaihe River floodplain, were found.

### 3.4. Analysis of Factors Affecting Waterbirds Habitat Importance

This study used a GAM to model the relationship between waterbird habitat importance, as assessed by field surveys. DEM, NDVI, area, and perimeter were significantly correlated with waterbird habitat importance (DEM: F_1.833, 1.972_ = 40.97, NDVI: F_1.95, 1.997_ = 22.97, area: F_1.988, 2_ = 141.1, perimeter: F_1.853, 1.979_ = 88.66; all *p* < 0.05), whereas road and people were not (road: F_1, 1_ = 0.97, *p* = 0.956; people: F_1, 1_ = 6.925, *p* = 0.783). DEM showed a significant negative correlation with waterbird habitat importance, while perimeter showed a positive correlation. Habitat importance had a bell-shaped relationship to both area and the NDVI, first increasing and then decreasing with increasing factor magnitude (Figure 6).

## 4. Discussion

### 4.1. Distribution of Important Habitats for Waterbirds

Our study revealed the distribution pattern of important habitats for waterbirds in Anhui Province and identified areas of current conservation gaps. We further analyzed the spatial and temporal differences in the distribution patterns of important habitats for waterbirds in the province. The results show that the number of important habitats increased significantly during the wintering period and that such habitats were mostly distributed in the Yangtze River floodplain. Seasonal changes in environmental factors such as water levels and surrounding vegetation in lakes, rivers, and reservoirs result in alterations in the availability of waterbird habitats, with varying impacts on waterbird communities. During autumn, water levels tend to decline and riparian habitats gradually diversify [31]. Low water levels during winter and spring are important for the emergence, growth, and reproduction of aquatic plants. During the dry season, increases in food resource abundance and availability have been observed in habitat areas such as mudflats and grass flats [32,33], enhancing the number of foraging sites available to waterbirds [34]. Moreover, the high water level in summer promotes the growth of submerged and water-supporting plants, which provide concealment for waterbirds during surveys. Consequently, the recorded number of waterbird species during wintering, spring migration, and autumn migration was higher than that during breeding, and the important habitats for waterbirds were mostly distributed at Shengjin, Caizi, and Huanghu Lake, where water levels were low. During breeding, waterbirds were mostly found in areas with richer vegetation, such as Huangpi, Daguan, and Wuchang Lake.

The hydrological fluctuations of the shallow lakes in the middle and lower reaches of the Yangtze River were similar to those of the river itself. The water level decreased significantly during the dry season. The aquatic habitats were mainly grassy beaches, mudflats, and shallow water, and the abundant food resources were highly accessible [35]. Wetland ranges with low levels of anthropogenic disturbance constitute desirable habitats for waterbirds. However, the Huaihe River floodplain mostly features reservoir or river-type wetlands with deep water habitats or wetlands of long and narrow shapes. Moreover, most wetlands in this area are subject to anthropogenic disturbances, such as purse seine fishery, land reclamation activities, and water level control changes caused by water conservancy projects, which also negatively affect habitat availability and accessibility. Such impacted habitats reduce the ability of waterbird communities to sustain themselves and may cause wintering waterbirds to experience reduced population sizes, ranges, and species richness [36]. Therefore, the largest number of important habitats and those with the greatest importance were located in the Yangtze River floodplain, such as Shengjin, Huangpi, Wuchang, and Caizi Lake, while the Huaihe River floodplain had relatively low numbers and levels of important habitats.

### 4.2. Waterbird Protection Gaps

Wetlands are one of the most important ecosystems on Earth and constitute essential breeding and wintering habitats for waterbirds. As transitional ecosystems between land and water, wetlands are subject to unique environmental conditions, and the composition of waterbird communities is closely related to that of specific wetland environments. The ongoing degradation and loss of wetlands requires urgent conservation measures, among which the establishment of protected areas is one of the most effective [37]. Owing to limited human resources, an efficient combination of scientific conservation planning and effective management is necessary for the successful implementation of such conservation efforts [38]. Currently, wetland conservation primarily depends on the creation of nature reserves, with national and provincial nature parks providing additional support. This framework plays a crucial role in wetland preservation. However, certain ecologically significant wetlands remain outside these formally protected areas. For instance, Huangpi Lake is an important wetland in China and receives some level of protection, and some areas around Chaohu Lake are designated as nature reserves, but the main body is not protected. Such cases underscore the need for enhancing protection, possibly through the establishment of additional nature reserves. Gap analysis is an effective tool for accomplishing this task [39]. Gap analysis is critical to biodiversity conservation at the regional scale, using the existing protected area system to identify areas that are not or insufficiently protected and prioritize conservation targets and actions. Adaptive hierarchical management in combination with wetland status assessment can be used in this approach to identify areas with a low or insufficient protection level.

### 4.3. Factors Affecting Waterbird Habitat Importance

Waterbirds are highly habitat-specific, while different habitat types attract different faunal communities [40]. Habitat characteristics strongly determine waterbird species abundance and richness in a given locality. These metrics are closely related to community structure and composition, and play important roles in maintaining waterbird community diversity [36]. Vegetation is an important driver, providing food resources and hiding places; however, excessive vegetation cover can negatively affect waterbird survival [35]. Wetland area and shoreline length are strongly determinative of the extent of habitat in the fallow zone, which, as a confluence of aquatic and terrestrial ecosystems, provides a high diversity of niches and abundant food resources [13,41]. We found that all three of these factors were positively correlated with the species number and abundance of waterbirds within a certain range. Wetlands at high elevations tended to be mountainous reservoirs with predominantly deep water habitats and few food resources and were thus inhabited by fewer waterbirds, leading to a negative correlation of elevation with waterbird habitat importance [42]. Human disturbance also affects waterbird communities, directly or indirectly producing long- and short-term behavioral, physiological, and reproductive changes [43,44]. Local human population density and the vicinity of roads were thus negatively related to habitat importance [7]. These findings demonstrate that differences in habitat characteristics and the presence of anthropogenic disturbances can alter habitat selection by waterbirds, resulting in waterbird habitats of varying importance.

## 5. Conclusions

This study established an index system for the evaluation of waterbird habitat importance based on habitat suitability indicators, assessed and ranked the importance of waterbird habitats in Anhui Province, and generated a ranked list and distribution patterns of important habitats for waterbirds in the region. Among the 306 waterbird habitats surveyed, 73 habitats were identified as important and were further analyzed to determine seasonal distributional changes. Elevation, vegetation cover, lake area, and lake perimeter length significantly affected habitat importance. Gap analysis revealed that 42 (57.53%) of these important waterbird habitats are not currently protected. Our findings provide valuable information for the protection of waterbird habitats in Anhui Province and demonstrate the application of a data-driven approach to prioritizing waterbird habitat protection measures.

As the extent of our dynamic analysis of waterbird distributions was limited in this study, further in-depth research on the spatial and temporal dynamics of waterbird habitat importance in the region should be conducted. The NDVI is primarily a measure of vegetation cover, while waterbird habitat selection may be influenced by the diversity and availability of plant resources. However, its effectiveness as an indicator for evaluating waterbird habitat importance is limited. A more comprehensive and accurate analysis should include a combination of multisource remote sensing data and field survey data on vegetation to better assess the influence of vegetation on the significance of wetlands for waterbirds.

## Figures and Tables

**Figure 1 animals-14-01004-f001:**
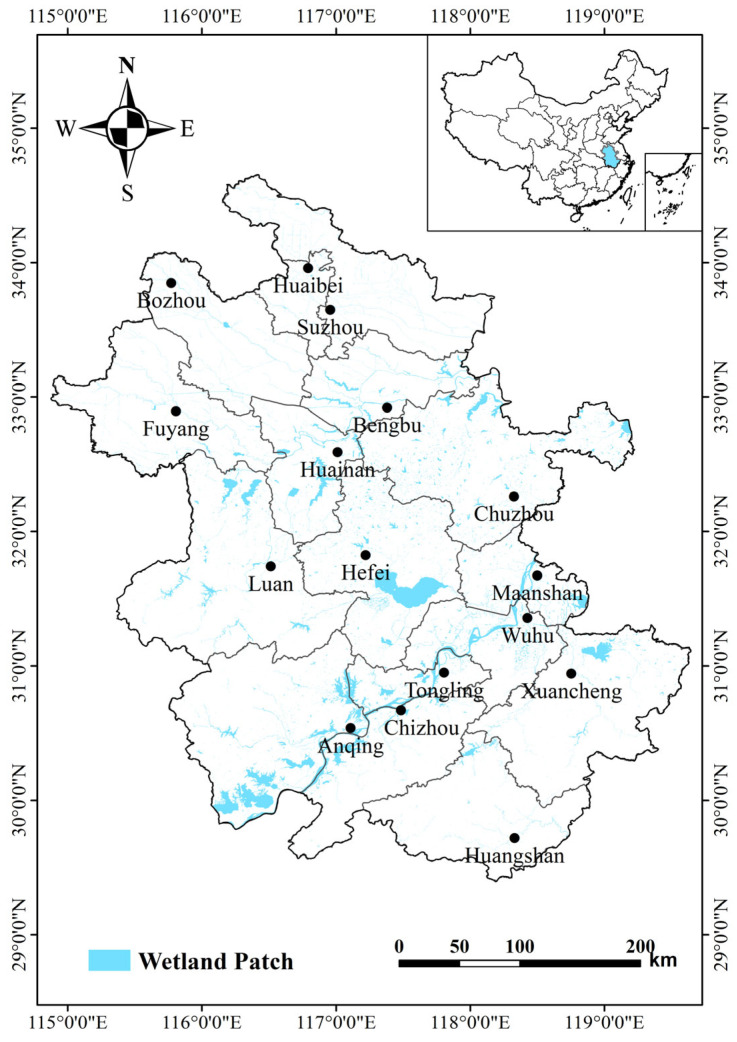
Study area in Anhui Province.

**Figure 2 animals-14-01004-f002:**
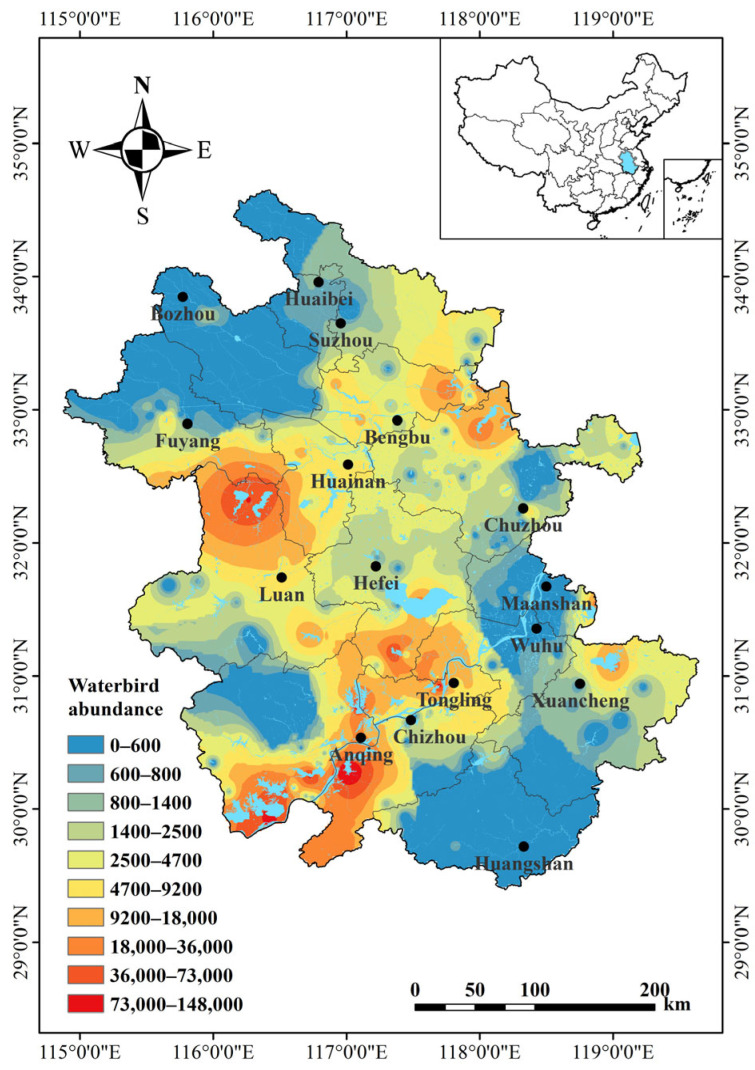
Distribution of waterbird populations in Anhui Province. The displayed range represents the total waterbird abundance (individual number) in each wetland in a year. Spatial interpolation of waterbird abundance data using inverse distance-weighted interpolation in ArcGIS to demonstrate the spatial distribution of species abundance.

**Figure 3 animals-14-01004-f003:**
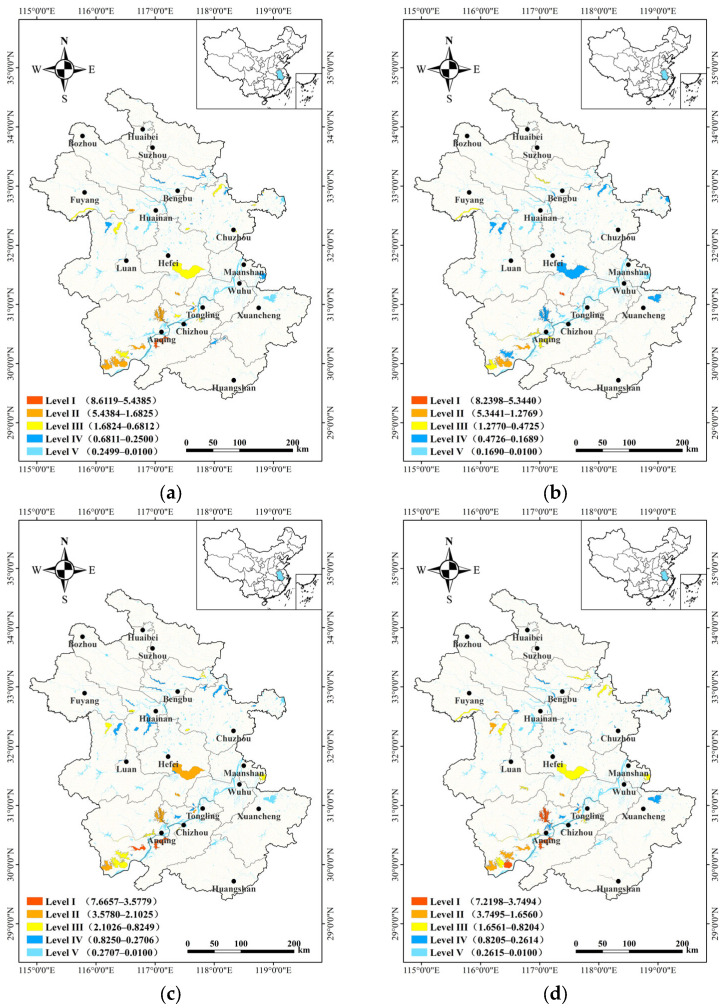
Important waterbirds habitats in Anhui Province during the four survey periods. Classification ranges from Level I (most important) to Level V (least important). Score ranges are given in parentheses. (**a**) Spring migration period; (**b**) breeding period; (**c**) autumn migration period; (**d**) wintering period.

**Figure 4 animals-14-01004-f004:**
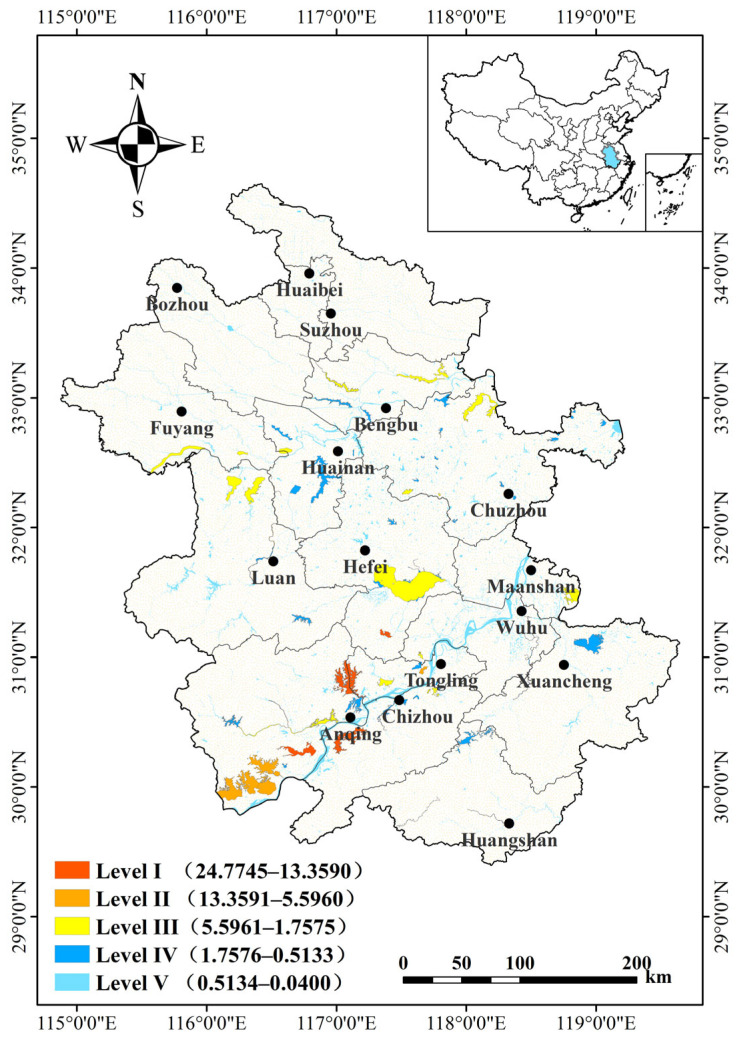
Important waterbird habitats in Anhui Province based on combined scores from all four seasonal surveys. Classification ranges from Level I (most important) to Level V (least important). Score ranges are given in parentheses.

**Figure 5 animals-14-01004-f005:**
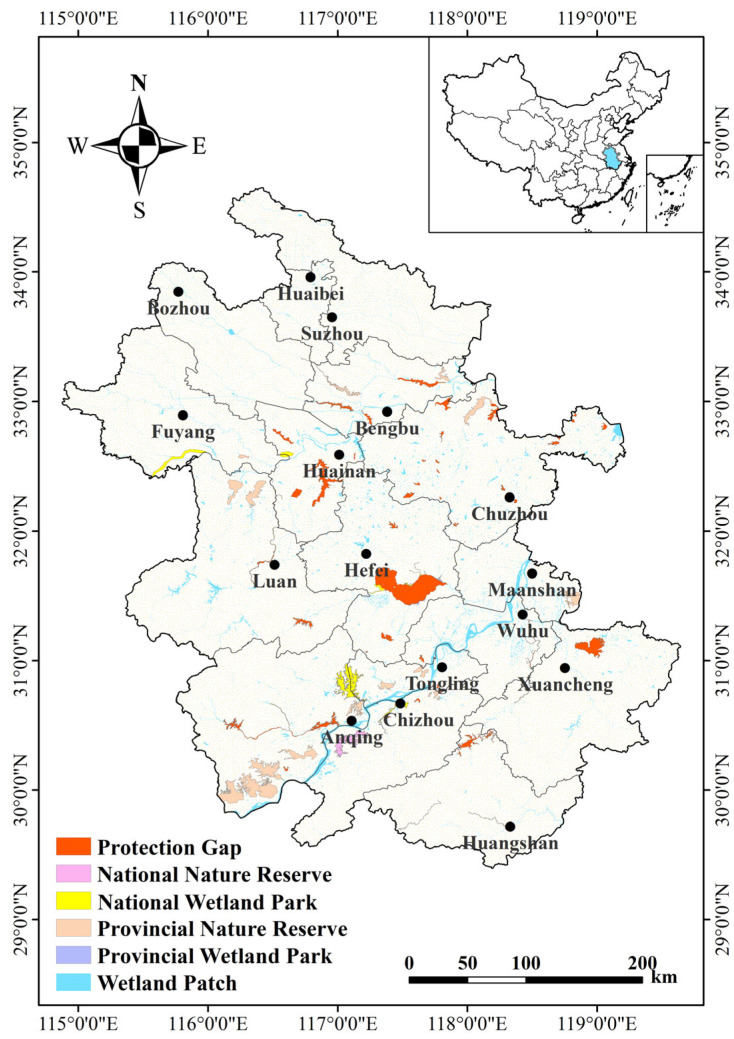
Waterbird habitat protection and identified protection gaps in Anhui Province. The scope of protection is national and provincial nature reserves and parks.

**Figure 6 animals-14-01004-f006:**
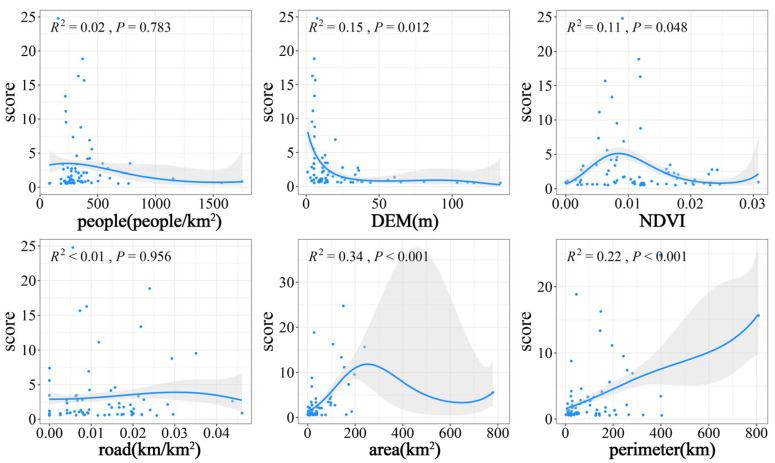
Relationship between habitat importance of waterbirds and environmental factors. The gray portion indicates a 95% confidence interval. The dots represent the locations and distributions of the observed data points and the lines represent the non-linear relationships between the variables and the response variables.

**Table 1 animals-14-01004-t001:** Evaluation index system of waterbird habitat importance in Anhui Province.

Aspect	Indicator	Description
Diversity	Waterbird Species Richness	The number of waterbird species recorded in the survey unit
	Waterbird Abundance	Estimate by multiplying waterbird density by area
	Diversity Index	Shannon-Wiener index
Importance	CR and EN Waterbird Species Richness	The number of CR and EN waterbird species recorded in the survey unit
	CR and EN Waterbird Species Abundance	The number of CR and EN waterbirds recorded in the survey unit
	VU Waterbird Species Richness	The number of VU waterbird species recorded in the survey unit
	VU Waterbird Species Abundance	The number of VU waterbirds recorded in the survey unit
	Cranes, Storks, and Pelicans Species Richness	The number of Cranes, storks, and pelicans species recorded in the survey unit
	Cranes, Storks, and Pelicans Species Abundance	The number of Cranes, storks, and pelicans recorded in the survey unit
	Ibises Species Richness	The number of Ibises species recorded in the survey unit
	National First-Level Protected Waterbird Species Richness	The number of National first-level protected waterbird species recorded in the survey unit
	National First-Level Protected Waterbird Species Abundance	The number of National first-level protected waterbirds recorded in the survey unit
	National Second-Level Protected Waterbird Species Richness	The number of National second-level protected waterbird species recorded in the survey unit
	National Second-Level Protected Waterbird Species Abundance	The number of National second-level protected waterbirds recorded in the survey unit
Suitability	Number of Waterbird Species with Waterbird Populations Exceed 1% of Flyway Population	
	Number of Waterbird Species with Waterbird Populations ≥ 1000	
	Number of Waterbird Species with Waterbird Populations ≥ 5000	

Note: CR = critically endangered, EN = near threatened, and VU = vulnerable on the IUCN Red List of Endangered Species. Nationally Protected Species from the National Forestry and Grassland Administration (https://www.forestry.gov.cn, accessed on 10 April 2023). Waterbird populations exceed 1% of flyway population means the percentage of the global or EAAF population published in Waterbird Population Estimates Fifth Edition (Ramsar Convention Secretariat, 2010; WPE5; Wetlands International, 2015; http://wpe.wetlands.org/, accessed on 10 April 2023).

**Table 2 animals-14-01004-t002:** Distribution of waterbirds in Anhui Province by the period in annual lifecycle.

Period	Waterbird Abundance (ind./Habitat)	Number of Species	NKPS Abundance (ind./Habitat)		IUCNRS Abundance (ind./Habitat)			
			First Class	Second Class	CR	EN	VU	NT
Spring migration	855.48 ± 174.69	0.23 ± 0.37	5.85 ± 2.23	104.71 ± 35.98	2.01 ± 0.85	3.11 ± 1.39	30.96 ± 14.17	133.23 ± 47.02
Breeding	187.89 ± 25.36	0.16 ± 0.26	0.17 ± 0.13	3.09 ± 1.44	0.17 ± 0.13	0 ± 0	0 ± 0	0.09 ± 0.04
Autumn migration	1280.07 ± 269.29	0.24 ± 0.36	3.6 ± 2.05	83.74 ± 38.89	2.03 ± 1.83	1.03 ± 0.54	21.16 ± 9.68	76.68 ± 26.56
Wintering	2450.66 ± 504.96	0.25 ± 0.44	17.66 ± 7.97	182.94 ± 59.13	1.12 ± 0.41	15.03 ± 6.77	54.3 ± 21.64	297.71 ± 129.17

Note: NKPS = National Key Protected Species, IUCNRS = IUCN Red List species, CR = critically endangered, EN = near threatened, VU = vulnerable, and NT = near threatened on the IUCN Red List of Endangered Species.

**Table 3 animals-14-01004-t003:** Distribution of waterbirds in Anhui Province by floodplain.

Floodplain	Waterbird Abundance (ind./Habitat)	Number of Species	NKPS Abundance (ind./Habitat)		IUCNRS Abundance (ind./Habitat)			
			First Class	Second Class	CR	EN	VU	NT
Yangtze River	7447.67 ± 2013.78	0.74 ± 2.52	62.06 ± 25.96	809.5 ± 288.28	12.52 ± 5.77	42.79 ± 19.22	132.88 ± 50.69	539.48 ± 185.27
Huaihe River	3235.38 ± 683.72	0.31 ± 1.36	3.33 ± 1.08	77.32 ± 23.39	0.34 ± 0.19	2.98 ± 1.07	98.89 ± 48.02	545.52 ± 251.05
Xin’an River	240.45 ± 96.5	2.1 ± 2.95	0 ± 0	11.05 ± 9.67	0 ± 0	0 ± 0	0.1 ± 0.1	5 ± 3.57

Note: NKPS = National Key Protected Species, IUCNRS = IUCN Red List species, CR = critically endangered, EN = near threatened, VU = vulnerable, and NT = near threatened on the IUCN Red List of Endangered Species.

## Data Availability

The data supporting the results of this article will be made available by the authors on request.

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
