# Peer review of "Assessment of Waterbird Habitat Importance and Identification of Conservation Gaps in Anhui Province"

_animals, 2024, doi:10.3390/ani14071004_

Round 1

Reviewer 1 Report

Comments and Suggestions for Authors

1. Drawing the distribution pattern of waterbirds in Anhui Province can more clearly reflect the distribution of waterbirds.

2. The study results are presented as mean values ( ± SD).

3. Please make sure that statistical test details are fully reported. You give p-values but not the test statistic.

4. L78-83 and L165-171: This section needs referencing.

5. L248-250 and L278-280: “R” and “P” should be italicized.

6. L246-248: It is recommended to change the sort to “, because of the degraded sort used in this article.

7.What is the reference of the bird classification standard adopted in this paper?  It is suggested to submit a list of birds involved in the results in the annex.

8. Other format problems and figures are suggested for further verification.

Comments on the Quality of English Language

Consider the language in the text, and rethink some of the words.

Reviewer 2 Report

Comments and Suggestions for Authors

I now have read the manuscript entitled "Assessment of waterbird habitat importance and identification of conservation gaps in Anhui Province" that was submitted to Animals. I comment the authors with an interesting work that must have been both difficult to accomplish and pleasant to perform. You will see that I have some minor remarks.

Overall, I find the introduction well organized and clear with the objectives well-presented. Also, the methods are well explained and seem appropriate to answer the aims of the project. The results are accessible and comprehensive with nice figures to illustrate them. Finally, the discussion is well structured, easy to follow and pleasant to read, but lacks a strong initial hook (see point-to-point remarks).

Otherwise, I have some point-to-point remarks:

L99-100. Where it reads: “Many waterbirds, including the endangered species Grus leucogeranus, Ciconia boyciana, and Aythya baeri, stop over each winter [24].”, the scientific names of the birds should be in italic.

L196-197. Where it reads: “Of these waterbirds, there were 122,940 species, (…)”, I believe the number of species is incorrect.

L284. In figure 5, it would be practical to add units to the x-axis of all the plots in order to assist reading them and their meaning.

L287. The beginning of the discussion lacks a strong initial hook summarizing the main results of this study and their most important implications. This is facultative, of course, but I believe the discussion becomes much stronger with such hook, locking in the reader with the most interesting information presented first, in a summarized manner.

Reviewer 3 Report

Comments and Suggestions for Authors

Wetlands are one of the most threatened ecosystems in the world. Determining which of them are priorities for waterbird conservation based on their environmental characteristics and importance for waterbirds is an essential tool for prioritising conservation actions. In this context, the use of indices for cataloguing their importance based on environmental parameters can help to identify those whose protection should be a priority, especially if they allow the identification of existing protection gaps in comparison with their environmental value. This study classifies the wetlands in Anhui Province according to a conservation index and compares this classification with a gap analysis of their protection status.

The study is very interesting and necessary to be able to prioritise conservation actions in those wetlands of high value for waterbirds in Anhui province. The objective and the methodology used is in principle adequate, cataloguing the importance of wetlands for waterbirds according to the evaluation of environmental variables and a subsequent comparison of their protection status by means of a gap mapping analysis. In that sense it is well thought out, executed and structured.

However, I have some theoretical rather than methodological doubts. I believe that the variables chosen may not be sufficient to determine the environmental importance of waterbird habitats and therefore their importance for conservation. The choice of lake area, and lake circumference gives us an idea of whether the size of the wetland is important, and of course it is, as highlighted. The main finding is that larger wetlands and with a higher NVDI are more used by waterfowl and therefore have greater importance in conservation. But I think you could have selected other environmental variables which are important to waterbirds: type of wetland, whether they are temporary or permanent, or whether they are natural or artificial, among others. In fact, in the article you argue that the more permanent or temporary nature of some wetlands is what makes them more attractive for waterfowl feeding. Could it then be that it is not the size of the wetland that is the main variable that makes it more attractive to waterfowl?

In this sense, I believe that a review of the results, and even of the analysis, would be more useful, trying to determine these variables, as this would further define which types of wetlands are more sensitive and need to be protected.

In any case, this is a suggestion, as the main result fulfils the objectives of finding the distribution of the most important wetlands for waterbirds in the province and through the gap analysis to define which wetlands are not protected and should be protected, providing a useful tool for land managers.

Specific suggestions for improving the text to clarify the review are detailed below.

Page 3 line 100. Scientific names should be in italics. Review all the text.

Page 3 line 101. I believe that reference number 24 refers exclusively to Anatidae, but to the other two species.

Page 6 line 196. I don’t understand “Of these waterbirds, there were 122,940 species,…”. What do you mean?

Reviewer 4 Report

Comments and Suggestions for Authors

The manuscript aims to 1) identify and rank important waterbird habitats in Anhui, China, 2) find protection gaps for important habitats for water birds, and 3) relate environmental factors to the importance of waterbird habitat. After reading the manuscript, I believe it is addressing an important issue, which is waterbird conservation, in a highly relevant region on the East Asia-Australia Flyway. With that being said, I think the authors can make the following changes to improve the manuscript:

1, (Line number 67 – 72): this paragraph is about a specific method, the entropy weights method. I would move it to the methods section and keep the introduction section conceptual.

2, (Line number 86 – 96): same issue as 1, I would move this paragraph to the methods section because it is strictly about a method (Gap analysis).

3, (Line number 97 – 107): this paragraph is entirely about the study site. I would move it to 2.1 study area.

4, (Line number 99 – 100): I would include good-quality photographs of those endangered waterbird species (G. leucogeranus, C. boyciana, and A. baeri). Doing so will better connect and engage the readers.

5, (Line number 125 – 129): I would expand a bit more about the study area. For example, some readers could be curious about the environmental and waterbird-related differences between wet and dry seasons.

6, (Line number 133 – 153): I would be more specific about the sampling design. In particular, I would talk about the number of observers who participated in the study and the method of randomization of the sampling effort. In addition, if the sample point method is the same as a point-count survey, then I would use “point count” or point-count survey”.

7, (Table 1): readers might not understand the reason behind including cranes, storks, and pelicans as part of the habitat importance evaluation. I would be explicit about this and clearly explain why these birds, among all other birds, are particularly relevant to waterbird conservation.

8, (The entire methods section): I would cite R and all the relevant R packages used for data analyses. Doing so will make this study more reproducible to future researchers.

9, (The entire discussion section): I would include more comparisons between this study and previous studies on the same topic, assuming there were relevant previous studies.

Comments on the Quality of English Language

As much as I value the good science in this manuscript, I believe the quality of English needs to be improved. I would recommend that the authors find a native speaker/editor to help with the English. Doing so will make the process of reading comprehension a lot smoother. Some issues related to Grammar and English that I have noticed include subject-verb agreement, overly long sentences, sentences without verbs, overuse of the word "notably", unclear pronoun references, and inaccurate word choices. One example of the last issue occurs on line 125. The way the word "distribute" is used is not correct. I would change the expression to "Anhui Province has ...".
